# New Therapeutics Targeting Arterial Media Calcification: Friend or Foe for Bone Mineralization?

**DOI:** 10.3390/metabo12040327

**Published:** 2022-04-05

**Authors:** Astrid Van den Branden, Anja Verhulst, Patrick C. D’Haese, Britt Opdebeeck

**Affiliations:** Laboratory of Pathophysiology, Department Biomedical Sciences, University of Antwerp, 2610 Antwerp, Belgium; astrid.vandenbranden@uantwerpen.be (A.V.d.B.); anja.verhulst@uantwerpen.be (A.V.); patrick.dhaese@uantwerpen.be (P.C.D.)

**Keywords:** arterial calcification, bone metabolism, cell death, oxidative stress, chronic kidney disease, phenotypic transition, nutrition, vascular therapy

## Abstract

The presence of arterial media calcification, a highly complex and multifactorial disease, puts patients at high risk for developing serious cardiovascular consequences and mortality. Despite the numerous insights into the mechanisms underlying this pathological mineralization process, there is still a lack of effective treatment therapies interfering with the calcification process in the vessel wall. Current anti-calcifying therapeutics may induce detrimental side effects at the level of the bone, as arterial media calcification is regulated in a molecular and cellular similar way as physiological bone mineralization. This especially is a complication in patients with chronic kidney disease and diabetes, who are the prime targets of this pathology, as they already suffer from a disturbed mineral and bone metabolism. This review outlines recent treatment strategies tackling arterial calcification, underlining their potential to influence the bone mineralization process, including targeting vascular cell transdifferentiation, calcification inhibitors and stimulators, vascular smooth muscle cell (VSMC) death and oxidative stress: are they a friend or foe? Furthermore, this review highlights nutritional additives and a targeted, local approach as alternative strategies to combat arterial media calcification. Paving a way for the development of effective and more precise therapeutic approaches without inducing osseous side effects is crucial for this highly prevalent and mortal disease.

## 1. Introduction

Enough is as good as a feast. A true story for mineralizing processes in the human body. Mineralization of bone and teeth is essential for their hardness and strength, whereas uncontrolled mineralization could lead to ectopic calcifications in extra-skeletal sites such as arteries and heart valves. The pathological buildup of calcium-phosphate crystals in the cardiovascular system occurs at four distinct sites: (i) atherosclerotic plaques or arterial intima calcification, (ii) media layer of the vessel wall or arterial media calcification, also known as Mönckeberg’s arteriosclerosis, (iii) heart valves or valvular calcification and (iv) small blood vessels in the skin or calciphylaxis. The focus of this paper will be on arterial media calcification, but we refer to excellent reviews on the therapeutic management of the other types of cardiovascular calcification [1,2,3]. Elderly patients and those suffering from dysmetabolic states, including diabetes and chronic kidney disease (CKD), either with or without the presence of osteoporosis, are at high risk to develop arterial media calcification. Furthermore, several genetically-mediated arterial diseases have been described as potential initiators of arterial media calcification, including pseudoxanthoma elasticum (PXE), arterial calcification caused by CD73 deficiency (ACDC), generalized arterial calcification of infancy (GACI) and Keutel syndrome [4,5]. The presence of arterial calcification poses an increased risk for cardiovascular disease and mortality by cause of reduced arterial compliance and arterial wall stiffness, which in turn precedes a multitude of severe cardiovascular consequences, including left ventricular hypertrophy, diastolic dysfunction and cardiac failure [6,7].

Despite the significantly high prevalence of arterial media calcification [8,9] and its tremendous economic burden to both patients and society [10], there still is a dearth of effective pharmacological therapies. Current anti-calcification treatments are confronted with a low efficacy because they do not directly tackle arterial media calcification but solely focus on amending common risk factors, such as hyperphosphatemia (e.g., phosphate binders). Likewise, these treatment strategies are restricted to patient populations with CKD as they aim to target CKD-specific conditions that are seldom seen without the occurrence of renal impairment. Moreover, treatments to date are confronted with limited therapeutic compliance due to important gastrointestinal side effects and also disturb physiological bone metabolism as arterial media calcification highly resembles bone mineralization [11,12]. This review presents a comprehensive overview of different anti-calcifying therapeutics and their impact on bone formation and sheds light on potentially novel targets. Continuous research is of utmost importance to develop efficient and safe therapies against calcification in the vessel wall without inducing osseous side effects, particularly for CKD, diabetes and osteoporosis patients since they often already suffer from a compromised bone quality.

## 2. An Overview of Pivotal Cellular and Molecular Mechanisms of Artery and Bone Mineralization

Arterial calcification resembles physiological bone mineralization, being a compromising factor for the development of anti-arterial calcification therapeutics, the more since CKD, osteoporotic and diabetic patients, who are the prime targets, also suffer from a deteriorated bone status. In this paragraph, we will discuss the main pathological events in arterial media calcification and this in relation to physiological bone mineralization. Emphasis is also put on whether the recent novel treatment strategies against arterial media calcification act as a friend or foe for physiological bone mineralization.

### 2.1. Targeting the High Phenotypic Plasticity in the Vasculature

The vascular wall consists of five main cell types: endothelial cells, vascular smooth muscle cells (VSMCs), pericytes, fibroblasts and vessel-residing stem cells. Arterial media calcification is a cell-mediated pathological process, predominantly driven by VSMCs [13]. These distinct cells have a high phenotypic plasticity to control local blood pressure (contractile phenotype) and repair the arterial wall after injury (synthetic phenotype). Interestingly, both VSMCs and osteoblasts originate from the mesenchymal stem cell. With this in mind, multiple in vitro/preclinical studies have shown that certain pathological triggers (i.e., high calcium and phosphate levels [14], uremic toxins [15], inflammation, oxidative stress [16]) induce a transition of VSMCs into cells with an osteo-/chondrogenic phenotype. Furthermore, patients with arterial media calcification show both intramembranous (without a cartilage intermediate) and endochondral (replacement of a cartilage intermediate into bone matrix) bone formation in the media layer of the vessel wall [17]. Similar to osteoblasts, these transdifferentiated VSMCs release and deposit extracellular matrix vesicles loaded with preformed calcium-phosphate crystals, enzymes, lipids and miRNAs into their extracellular matrix [18,19]. The transdifferentiation of VSMCs to cells with a bone-forming phenotype goes along with an upregulation of osteo-/chondrogenic marker genes, including runt-related transcription factor 2 (Runx2), alkaline phosphatase (Alpl), secreted phosphoprotein 1 (Spp1) and bone gamma-carboxyglutamate protein 2 (Bglap2) [13]. Recent studies have shown that epigenetic regulator miRNA-103a and sirtuin-6, as well as an SGK1 (serum- and glucocorticoid-inducible kinase 1) inhibitor, prevented VSMC calcification by inhibiting upregulation of bone-marker genes Runx2 and Msh homeobox 2 (Msx2) [20,21,22], thereby putting forward VSMC transdifferentiation as a valuable target to treat arterial media calcification. Furthermore, arterial stiffness, a well-known consequence of arterial media calcification, favors bone-like switching of VSMCs by facilitating the nuclear translocation of mechanical stimuli sensors Yes-associated protein (YAP) and its highly related transcriptional co-activator with PDZ-binding motif (TAZ). Subsequently, nuclear translocation of YAP/TAZ results in an upregulation of the mRNA expression of Runx2, Alpl, Spp1 and SRY-Box Transcription Factor 9 (Sox9) in the VSMC [23,24]. Another theory to block arterial calcification is stirring the VSMC transdifferentiation toward an adipocyte phenotype instead of an osteo-/chondrogenic phenotype. Adipocytes also originate from mesenchymal stem cells. Interestingly, the adipogenesis regulator sclerostin is suggested to halt VSMC calcification by suppressing the Wnt/b-catenin signaling cascade [25]. It is known that Wnt/b-catenin signaling in bone cells favors Runx2 expression while suppressing adipogenic differentiation [26,27]. Given the striking similarities between bone-forming cells and transdifferentiated VSMCs, the above-described therapeutic approaches have been linked to interference with bone formation [28,29]. Although, the extent of osteo-/chondrogenic marker gene (i.e., Alpl, Spp1, Bglap2) expression in transdifferentiated VSMCs is 40-fold lower compared to that in osteoblasts [30]. For this reason, it will be imperative to check whether dosages of these therapeutics targeting VSMC transdifferentiation can be administered so that only calcification in the vasculature is affected, while keeping bone mineralization intact.

Another important cell type in the vascular wall are the endothelial cells which also possess a high phenotypic plasticity known as the endothelial to mesenchymal transition (EndMT). This phenomenon acquires endothelial cells with a multiple differentiation potential toward fibroblasts/myofibroblasts, osteoblasts/osteocytes, chondrocytes and adipocytes [31]. Multiple in vitro and in vivo studies have shown the involvement of EndMT in arterial calcification [32,33,34]. However, EndMT is mainly regulated by transforming growth factor β (TGFβ)/bone morphogenic protein (BMP) signaling, which also plays a crucial role in osteoblast differentiation and mineralization by favoring transcription of Runx2 [35,36]. Targeting EndMT in arterial calcification seems to be attractive, but again, a close eye has to be kept on physiological bone mineralization.

Lastly, the outer layer of the arterial wall, also called the adventitia, is housed by vessel-residing stem cells, including Gli1+ cells or VSMC progenitors. A trigger, such as chronic renal failure, may induce Gli1+ migration toward the intima and medial layer of the vessel wall, followed by osteo-/chondrogenic transdifferentiation [37]. Also, pericytes, observed in all layers of the arterial wall are suggested to be a type of mesenchymal stem cells [38,39]. Pericytes are able to differentiate into osteoblasts, chondrocytes or adipocytes, depending on their trigger [40,41]. Additionally, pericytes also act as macrophage precursors in the brain, making them interesting therapeutic targets for arterial calcification treatment as they might facilitate a ‘cleanup’ of calcium-phosphate crystals in the calcified artery [42,43]. However, more research is needed to further characterize the phenotypic switching of vessel-residing stem cells and pericytes, in particular with regard to their expression levels of bone-like marker genes versus osteoblasts, migration status and degree of dominance in the arterial calcification process compared to VSMCs and endothelial cells.

### 2.2. Targeting Circulating Calcification Inhibitors and Stimulators

During arterial calcification, passive precipitation of saturated serum levels of calcium and phosphate also occurs. However, our body produces several calcification inhibitors including fetuin-A, pyrophosphate, matrix Gla protein (MGP) and osteopontin to avoid this calcium-phosphate crystal precipitation [44]. Unfortunately, in the presence of particular disease states, such as CKD, diabetes, osteoporosis or monogenic forms of arterial calcification, circulating calcification stimulators (including inflammatory mediators, uremic toxins, high phosphate and/or calcium levels, high glucose levels, oxidative stress factors) take the upper hand [44]. In the context of CKD, patients are routinely given phosphate binders, vitamin D analogs and calcimimetics to restore the imbalance in calcium and phosphate serum levels [45]. However, with regard to arterial calcification, these therapies are prone to low efficacy and compliance and are restricted to CKD patients only as hyperphosphatemia and –calcemia are rarely seen in individuals with normal renal function (osteoporosis patients and patients with monogenic forms of arterial calcification).

Multiple studies have focused on increasing calcification inhibitors to tackle the calcification process in the vessel wall. Pyrophosphate is a well-known calcification inhibitor as it prevents the incorporation of inorganic phosphate into hydroxyapatite crystals [46]. Oral administration of pyrophosphate was believed not to be a suitable treatment strategy due to its hydrolysis by intestinal alkaline phosphatases. However, researchers have shown that supplementing drinking water with 0.3 mM of pyrophosphate resulted into a significant reduction of tissue calcification in mice suffering from monogenic forms of arterial/connective tissue calcification (PXE and GACI) [47]. Moreover, clinical trials are running in which pyrophosphate (ClinicalTrials.gov Identifier: NCT04868578) is orally administered to PXE patients. Furthermore, tissue non-specific alkaline phosphatase (TNAP), expressed by calcified VSMCs, mediates the hydrolysis of pyrophosphate into inorganic phosphate [48]. Recently, our research group provided evidence for the TNAP-inhibitor SBI-425 to be able to inhibit warfarin-induced arterial media calcification in rats which, however, went along with bone mineralization side effects [49]. This was rather unexpected as (i) basal TNAP levels are 100-fold higher in osteoblasts compared to VSMCs [50], thus suggesting that the dosage of TNAP inhibitor needed would have minor effects on osteoblasts and (ii) studies performed on mice did not show any side effects on bone mineralization. However, it needs to be pointed out that it is more reliable to measure bone metabolism in rats versus mice due to (i) the small size of the bone area being measured in mice and (ii) the bone status of rats being more closely related to humans [51]. Furthermore, TNAP-inhibitor SBI-425 failed to stop the development of more severe arterial calcifications in an adenine-induced CKD rat model [52]. Interestingly, it has been shown by in vitro experiments that halting VSMC calcification goes along with an upregulation of TNAP activity, while an opposite effect has been seen during osteoblast mineralization [50]. In addition, TNAP only regulates 50% of the pyrophosphate hydrolysis, suggesting that other alkaline phosphatases may be more important in the CKD-induced vascular calcification process [53].

The calcification inhibitor fetuin-A interacts with pre-formed calcium-phosphate minerals creating calciprotein particles. These particles are cleared from the circulation via the reticuloendothelial system. However, during arterial calcification, these calciprotein nanoparticles undergo re-arrangement into more densely packed needle-shaped particles and precipitate in the arteries [54,55]. Moreover, low circulating levels of fetuin-A are linked to high calcification scores in CKD patients on dialysis [56]. Recently, an interesting drug SNF472 has been used to target coronary artery calcification (phase 2 clinical trials) and calciphylaxis (phase 3 clinical trial) in end-stage renal disease patients on hemodialysis [57,58,59,60]. SNF472 is an intravenous formulation of myo-inositol hexakisphosphate (IP6), a natural phytate product. This compound is a promising therapeutic as it targets the growth and formation of solid calcium deposits (hydroxyapatite) while not affecting free calcium, thereby avoiding the risk for hypocalcemia and exerting a therapeutic efficacy independent of the etiology of arterial calcification [61]. However, an important drawback of SNF472 is its short plasma half-life and modest potency, which is circumvented by injecting this drug through the dialysis machine during the hemodialysis sessions. This limits the use of SNF472 for CKD patients stage 3–4 not undergoing dialysis but already developing cardiovascular calcifications. For this reason, the research group of Schantl et al., is designing more pharmacological stable derivatives of IP6 (i.e., (OEG2)2-IP4) [62]. With regard to the bone, however, an imbalance between osteoid deposition and subsequent bone mineralization was observed in (OEG2)2-IP4-treated CKD rats [62]. In addition, the highest intravenous dosage of SNF472 (600 mg) reduced bone mineral density in end-stage renal disease patients with coronary artery calcification [63], again pointing out the need for caution with anti-arterial calcification therapeutics in target populations with an already compromised bone status.

Lastly, we describe the group of vitamin K-dependent calcification inhibitors. Vitamin K is needed for the gamma-carboxylation or activation of MGP, growth arrest-specific 6 (Gas6) and gla-containing coagulation factors, e.g., prothrombin [64,65,66]. Vitamin K deficiency is a dominant feature in the CKD population and thus also a well-established risk factor for arterial calcification [67,68]. For example, vitamin K-dependent gamma-carboxylation of MGP prevents arterial calcification through interfering with the binding of BMP2 to its receptor and by this inhibiting osteo-/chondrogenic transdifferentiation of VSMCs [69]. Recent studies have also shown that coagulation might play an important role in the arterial calcification process. Long-term exposure to protein-bound uremic toxins induced calcification in the aorta of CKD rats and was associated with the upregulation of coagulation pathways (i.e., extrinsic/intrinsic prothrombin activation pathway) [70]. This was in line with the results of Kapustin et al., revealing that Gla-containing coagulation factors prothrombin, protein C and S inhibited VSMC calcification [66]. Furthermore, Gas6, another vitamin K-dependent calcification inhibitor, exerts its anti-arterial calcification effects by preventing endothelial cells and VSMCs to go into apoptosis [71]. The effect of cell death, in relation to vitamin K, in the arterial media calcification process will be discussed in the next paragraph. Taken together, restoring the vitamin K status in CKD patients would be a valid anti-arterial calcification therapy. However, a recent multicenter randomized controlled trial showed that withdrawal of vitamin K antagonists in hemodialysis patients did not influence the progression of arterial calcification after 18 months [71]. On the other hand, there is still some debate on whether or not it is advisable/effective to administer vitamin K supplements (in the form of menaquinone-4 and 7) to CKD and diabetes patients [72,73,74]. Larger randomized clinical studies, as well as longer observations, are needed to reveal the anti-arterial calcification effects of vitamin K supplementation.

### 2.3. Targeting Cell Death Events in the Vasculature

Next to vascular cell transdifferentiation and a disbalance between pro- and anti-calcifying factors, VSMC death is a central process in the onset of calcification in the vasculature. Interestingly, VSMC death is boosted in pediatric patients undergoing hemodialysis and the initiation of hemodialysis treatment is associated with shifting the onset of arterial calcification, which begins predialysis, into overt calcification [75]. Also, cultured human VSMCs exposed to serum from uremic patients displayed extensive VSMC death [76]. VSMC-derived apoptotic bodies, enriched with high calcium concentrations, are released in the extracellular matrix and act as an excellent nidus for the deposition of the crystals [77,78]. Cellular DNA is released as well upon VSMC death, which has been demonstrated to initiate arterial calcification by precipitating calcium-phosphate crystals in the vessel wall [79]. Even though (i) Proudfoot and her team already suggested more than twenty years ago that VSMC death significantly contributes to the development of arterial media calcification [77,78] and (ii) Patel et al., showed that VSMC calcification was associated with a 50% increase of apoptosis while osteoblast’s viability remained unchanged [30], no effective treatments targeting VSMC death without imposing detrimental effects on physiological bone formation has been developed yet. For example, the caspase inhibitor ZVAD.fmk inhibited calcification in human VSMC nodules via antiapoptotic effects [77] and caspase inhibition was also demonstrated to inhibit the maturation and release of apoptotic bodies [80]. Since caspases also execute non-apoptotic roles involved in osteogenesis [81,82] and pan-caspase inhibition resulted in significant alterations in the expression of several osteogenic genes [81,83], effects of these possible anti-arterial calcification treatments at the level of the bone need further investigation. In addition to preventing apoptotic cell death, caspase inhibitors are shown to induce a switch from apoptosis to necrosis [84,85]. Interestingly, necrotic cell death is a well-known hallmark of atherosclerosis [86,87,88]. Apoptotic bodies, characterized by a preserved integrity of their membrane, are rapidly and efficiently phagocytosed. However, impaired clearance of apoptotic bodies results in secondary necrosis inducing membrane rupture and thus leakage of calcium and phosphate [87,88]. As the role of necrosis in arterial media calcification is not yet clear, one should be cautious with using caspase inhibitors since this may lead to necrotic cell death and trigger calcification.

The absence of effective treatments targeting VSMC apoptosis as indicated by the substantially unspecific TUNEL assay (i.e., apoptosis detection method), which is considered to detect DNA damage in general and thus also detects non-apoptotic forms of cell death [89], supports the involvement of other cell death types in arterial calcification. This deserves further interest, especially since several new types of regulated cell death have been identified during recent years [90]. Ferroptosis is an iron-dependent form of regulated cell death in which an excess of iron, mainly ferrous iron, via the Fenton reaction induces the generation of free radicals. This in turn initiates lipid peroxidation of polyunsaturated fatty acid-containing phospholipids, ultimately resulting in the accumulation of toxic lipid hydroperoxides and membrane damage [91,92]. Because the contribution of lipid peroxidation and ferroptosis to vascular diseases, such as atherosclerosis [93,94], but not yet to arterial media calcification, is a well-known phenomenon, the latter should be considered. The administration of vitamin E (i.e., membranous lipid peroxidation inhibitor) and selenium (i.e., important co-factor for glutathione peroxidase 4 which detoxifies lipid hydroperoxides) has already been shown to inhibit calcification in cultured VSMCs [95,96]. Furthermore, iron tends to be sequestered in vascular cells of CKD patients as a result of hepcidin upregulation, which degrades the iron exporter ferroportin, and therefore accelerates the production of hydroxyl radicals [97,98]. Interestingly, hemodialysis patients are administered iron intravenously to alleviate their functional iron deficiency, further favoring cellular iron sequestration. A direct calcification aggravating effect of iron seen in vitro [99] further substantiates the potential role of iron sequestration and subsequent lipid peroxidation/ferroptosis.

The use of lipophilic membranous radical trapping agents (e.g., vitamin E and ferrostatin-1) is a possible way to tackle lipid peroxidation/ferroptosis. Fascinatingly, vitamin E exerts bone-protecting functions [100] and in a study by Valanezhad et al., the administration of ferrostatin-1 to the MC3T3-E1 osteoblast cell line, in which ferroptosis was induced, promoted osteoblast differentiation [101]. Also, treatment with the anti-diabetic drug metformin attenuated calcifying and ferroptotic events in VSMCs and rats fed a high-fat diet [102]. Besides, metformin shows stimulatory effects on bone formation by promoting osteoblastic differentiation partly via AMP-activated protein kinase (AMPK) signaling pathway activation [103,104]. Melatonin has previously been published to reduce high-glucose-induced ferroptosis in osteoblasts in an in vitro and in vivo model of type 2 diabetes mellitus via activation of the Nrf2 signaling pathway [105]. Interestingly, (i) stimulating Nrf2 activity has already been suggested to suppress arterial calcification by regulating VSMC death and osteogenic transdifferentiation [106], (ii) melatonin attenuates calcification of cultured VSMCs [107] and (iii) clinical trials investigating the effect of melatonin on coronary artery calcification are running (ClinicalTrials.gov Identifier: NCT03966235 and NCT03967366). This, together with melatonin’s ability to enhance bone formation and improve osteoporotic lesions [108,109,110], puts forward melatonin supplementation as a possible therapeutic strategy for arterial calcification with a win-win situation for patients with a disturbed bone metabolism.

Preventing an excess of labile redox-active iron and cellular iron sequestration can also be hypothesized to be beneficial in the pathological process of arterial calcification as well as disturbed bone metabolism. Iron overload is known to influence bone formation by inhibiting osteoblast proliferation and differentiation and facilitating osteoclast differentiation [111,112]. Treatment with lactoferrin, an iron-chelator, improved bone formation and reduced bone resorption [113]. However, iron has been controversially published to suppress arterial calcification in the vessel wall of CKD-induced rats by the prevention of osteo-/chondrogenic transdifferentiation of VSMCs and suppression of the CKD-induced elevated expression of the phosphate transporter Pit-1 [114]. This might be linked to the fact that iron supplementation to CKD patients in first instance restores their functional iron deficiency and thereby prevents further renal function decline: our research group demonstrated that intestinal uptake of iron in CKD rats significantly reduced arterial calcification in combination with a better preservation of renal function [115]. Figure 1 gives a schematic overview of VSMC death-related processes and their interference with the bone mineralization process.

### 2.4. Targeting Oxidative Stress in the Vasculature

Oxidative stress, manifested by the production of reactive oxygen species (ROS), drives the progression of arterial media calcification by regulating VSMC death and the osteo-/chondrogenic phenotypic switch [116], as visualized in Figure 1. The prevention of oxidative stress may thus tackle both VSMC death and the osteo-/chondrogenic transdifferentiation of VSMCs. The production of ROS is normally encountered by antioxidant enzymes but pathophysiological settings trigger excess production of these highly reactive molecules and consequently imbalanced ROS homeostasis [117]. This opens the opportunity for antioxidant therapy to become a potential candidate to counteract oxidative stress and subsequently arterial calcification. Restoring the balance in oxidant formation and antioxidant capacity might be beneficial for a disturbed bone metabolism as well because oxidative stress induces apoptosis of osteoblasts and osteocytes and underlies the differentiation of pre-osteoclasts into osteoclasts [118,119]. Whereas several natural antioxidants from dietary sources, e.g., quercetin [120] and diosgenin [121], or non-dietary sources, e.g., rosmarinic acid [122] and synthetic compounds (e.g., sodium thiosulfate [123,124] and hydrogen sulfide [125]) showed anti-calcification properties both in vitro and in vivo [126], administration of other anti-oxidative agents resulted in controversial effects. Tempol on one hand ameliorated arterial calcification in a CKD rat model via the reduction of ROS in a study of Yamada et al. [127], while on the other hand Bassi et al., showed increased medial calcification as a result of tempol administration [128]. Although anti-oxidants positively affect bone homeostasis by activating osteoblast differentiation and reducing osteoclast activity [119], the use of anti-oxidants as potential anti-calcific treatment should be further studied with a special focus on the relationship with physiological bone metabolism. Furthermore, in response to oxidative stress and DNA damage, poly(ADP-ribose) [PAR] is synthesized by PAR polymerases (PARP) to favor DNA repair [129]. However, PAR, which is released in the extracellular matrix, promotes the pathological arterial calcification process by stimulating the osteogenic transition of VSMCs [130]. The antibiotic minocycline, a PARP inhibitor, reduced the development of arterial media calcification in adenine-induced CKD rats. Unfortunately, minocycline’s anti-arterial calcification effects went along with a decrease in cortical thickness and mineral density in the long limb bones of the animals [131]. Indeed, another PARP inhibitor, PJ34, impeded osteogenic metabolism by regulation of BMP-2 signaling [132].

## 3. Nutritional Care to Treat Arterial Media Calcification

Furthermore, nutritional care is essential to slow down the progression of CKD and diabetes but could also be beneficial for halting the development of arterial calcification. Food additives, including quercetin [133,134], curcumin [135,136], vitamin K [137] and phytates [138] increase the activity of calcification inhibitors and reduce oxidative stress in the arterial wall [139]. Next to this, it has been shown that vitamin K supplements inhibit arterial calcification [137] while stimulating bone mineralization [140,141]. Similarly, melatonin protects against calcification in the vessel wall but also improves osteoporotic lesions [110]. Also, magnesium supplementation has been demonstrated to effectively target arterial calcification by passively binding inorganic phosphate, in this way reducing the formation of hydroxyapatite crystals, and actively targeting VSMC transdifferentiation and VSMC death [142,143]. In a study of Diaz-Tocados et al., moderate (0.3%) dietary magnesium supplementation to uremic rats significantly reduced arterial calcification whilst improving bone metabolism [144] and in humans, magnesium has been shown to effectively prevent the progression of arterial calcification [145,146]. Although magnesium plays a key role in bone health, the association between magnesium supplementation and improving bone health requires further in-depth research [147]. Furthermore, eicosapentaenoic acid (EPA), an omega-3 polyunsaturated fatty acid that can find its source in fatty fish and its fish oils, is reported to directly inhibit arterial calcification [148,149,150], while improving osteoporotic bone status by the inhibition of osteoclast activity [151,152]. These types of anti-arterial calcification therapeutics could be especially beneficial to patients experiencing a low bone turnover status, such as osteoporosis patients and a substantial group of CKD patients. However as mentioned above, a single therapy strategy (i.e., vitamin K supplementation) will be not effective since the arterial calcification process is a result of a complex interplay between different pathological pathways and thus indicating the need for a multifactorial treatment strategy. On the other hand, improving imaging techniques to catch microcalcifications (PET scan 18-FDG), allowing start of treatment at an early stage, might allow the use of low dosages of anti-arterial calcification compounds and by this reduce side effects on the bone [153].

## 4. Towards a Local Approach to Tackle Arterial Media Calcification

Nanomedicine is an upcoming field with many therapeutic applications found mainly in cancer and other diseases. Multiple nanoparticle-based drug delivery systems have been approved by the FDA. The great advantage of nanomedicine is the ability of conjugating specific proteins on the surface of the nanoparticle, as well as shielding of compounds from circulating degrading enzymes leading to higher bioavailability and prolonged blood circulation. Interestingly, the research group of Vyavahare et al., developed a nanoparticle conjugated to an elastin antibody to target the diseased vasculature [154]. Elastin degradation is a typical feature in arterial calcification as VSMCs express matrix metalloproteinases cleaving the elastin fibers [155,156]. Vyavahare et al., showed that targeted delivery of albumin nanoparticles loaded with a calcium-chelating agent (EDTA) and conjugated to an anti-elastin antibody blocked the development and progression of arterial media calcification in CKD rats without inducing side effects on bone mineralization [157]. Moreover, this nanoparticle-based targeted delivery has also shown its efficacy in in vivo models for abdominal aortic aneurysm which is also characterized by degraded elastin fibers [158,159,160]. Nonetheless, therapeutic efficacy of nanomedicine highly depends on a number of factors, such as nanoparticle size, charge and distribution. These parameters influence the in vivo fate of the nanoparticles, including systemic distribution, cellular uptake and circulation lifetime. In addition, cost-effectiveness balance can be tricky [161]. High amounts of drugs could be necessary to obtain a significant percentage of drug entrapment into the nanoparticles. In order to optimize therapeutic efficiency, it is indispensable to conduct numerous trial and error procedures for optimizing the nanoparticle preparation parameters.

## 5. Extrapolation of Anti-Arterial Media Calcification Therapeutics toward the Human Situation and Other Types of Cardiovascular Calcification

Many therapeutic approaches against arterial media calcification are still in the pre-clinical phase. However, to which extent can the findings in animals be translated to the human situation? With regard to arterial media calcification, rodent models are primarily used. For example, (i) non-CKD models, such as genetically modified mice expressing less calcification inhibitors (i.e., fetuin-A, MGP), and rats receiving high dosages of warfarin and (ii) CKD-related models, including rats undergoing 5/6th nephrectomy or receiving an adenine diet, both combined with high phosphate intake [162,163]. Typical features of human arterial media calcification can be found in these animal models: VSMC transdifferentiation, low levels of circulating calcification inhibitors, VSCM cell death and oxidative stress. Also, the prevalence of CKD-induced arterial media calcification (40–70%) in these animals models is comparable to the human situation [17,164,165,166]. At the moment, larger animal models (i.e., pigs, rabbits, dogs) for arterial media calcification are lacking, which is unfortunate as the cardiovascular anatomy and physiology of larger animals are more comparable to the human situation. On the other hand, zebrafish are increasingly used to study cardiovascular disease as alternatives for non-human primates, pigs and rodents [167]. Despite the relative simplicity of their cardiovascular system, the heart rate and vascular anatomy are highly comparable to those of humans, and they share many (pathological) mineralization-related pathways [167,168]. Although the blood pressure is an important difference, zebrafish show similar responses to vasodilators, vasoconstrictors and cardiovascular drugs (e.g., nitric oxide donor sodium nitroprusside) [167,169]. Next to this, knock-out of specific calcification-related genes can be performed by the injection of morpholinos [170,171], and zebrafish have already been used to study calcification in PXE and GACI [172,173,174], further favoring the use of zebrafish as an interesting alternative animal model to study arterial media calcification. Additional advantages are the transparency of the embryos, allowing non-invasive observation of the blood vessels, and their short life span, making it an attractive model to study age-related diseases [167].

Some of the above mentioned treatment strategies could also be promising therapies for the other types of cardiovascular calcification. Arterial media calcification shares similar pathological mechanisms to arterial intima calcification and valvular calcification; i.e., precipitation and nuclear growth of calcium-phosphate crystals, transdifferentiation of vascular cells into osteo-/chondrogenic like cells, imbalance in circulating calcification inhibitors and stimulators, cell death and oxidative stress [175]. However, there is still debate as to whether it is beneficial to block atherosclerotic plaque calcification. Highly calcified plaques may be considered as stable atherosclerotic plaques, while less or spotty calcified plaques are associated with plaque rupture [176].

## 6. Conclusions

In conclusion, arterial media calcification is (i) a complex, multifactorial disease and (ii) very challenging to tackle due to its similarities to physiological bone mineralization. Therefore, in the future a focus has to be put on combination therapies to interfere with the multiple key mechanisms of arterial media calcification and target therapies directly to the diseased vasculature (i.e., by using nanoparticles) to avoid compromising the bone compartment. Nutritional additives have been shown to exert beneficial effects in halting the progression of CKD and the development of arterial media calcification and can therefore complementarily contribute to this multifactorial treatment strategy. A first considerable group of anti-calcification therapeutics includes targeting vascular transdifferentiation during which special focus needs to be paid to selecting the appropriate doses in order to prevent adverse effects on physiological bone metabolism. Next, restoring the balance between calcification inducers and inhibitors can be considered. Finally, as VSMC death plays a pivotal role in the pathological process of vascular calcification, focusing on targeting cell death, especially ferroptosis, is a promising therapy. The ferroptotic role in bone metabolism, however, has not yet been elucidated, requiring in-depth research before assuming that targeting lipid peroxidation/ferroptosis to tackle arterial media calcification does not affect physiological bone mineralization. In the development of anti-calcifying therapeutics, animal models are key, but the translation to the human situations remains challenging and requires critical evaluation.

## Figures and Tables

**Figure 1 metabolites-12-00327-f001:**
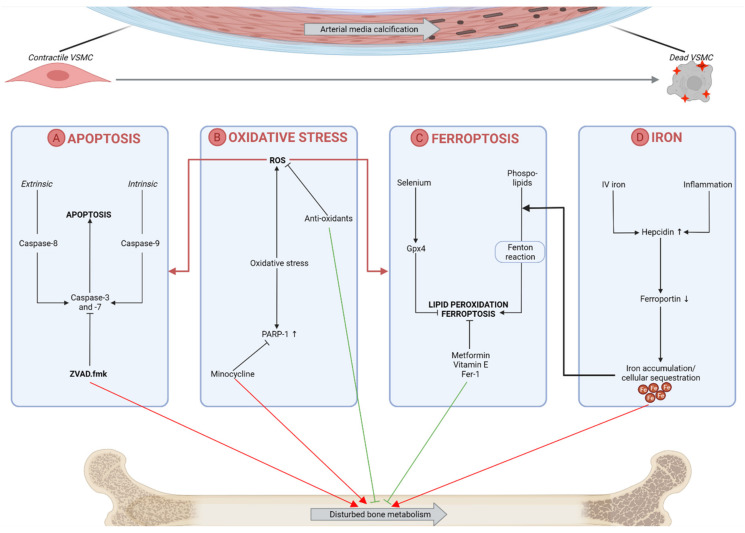
Targeting vascular smooth muscle cell (VSMC) death and oxidative stress-related processes as a possible way to tackle arterial media calcification and its effect on bone mineralization. (**A**) Apoptosis, a caspase-dependent type of cell death, contributes to the development of arterial media calcification. Apoptotic bodies act as a nucleation site for the deposition of calcium-phosphate crystals. The caspase inhibitor ZVAD.fmk has shown its efficiency in inhibiting arterial calcification but may cause detrimental effects on bone metabolism as caspases play an important role in physiological bone mineralization as well. (**B**) Oxidative stress, a central process in the onset of arterial calcification, either results in the generation of reactive oxygen species (ROS) which in turn drives the onset of apoptosis and ferroptosis, or the upregulation of poly(ADP-ribose) polymerase-1 (PARP-1) which stimulates the osteogenic transdifferentiation of VSMCs. Antioxidants and PARP inhibitors (i.e., minocycline) encounter arterial calcification but respectively enhance or disturb physiological bone mineralization. (**C**) A possible role of lipid peroxidation and ferroptosis (i.e., an iron-mediated type of regulated cell death) in arterial media calcification. Both mechanisms might be targeted without causing side-effects on the bone metabolism as metformin, vitamin E and ferrostatin-1 (Fer-1), which are therapeutics shown to inhibit ferroptotic events, show stimulatory effects on bone formation. Furthermore, selenium administration is known to inhibit VSMC calcification but also is an important co-factor of glutathione peroxidase 4 (Gpx4), an important regulator in ferroptosis. (**D**) The intravenous (IV) injections of iron to CKD patients on dialysis and inflammatory states cause a hepcidin upregulation, which in turn downgrades ferroportin (i.e., an important iron exporter) and causes intracellular iron sequestration. On one hand, this can induce lipid peroxidation and ferroptosis, and on the other hand, iron overload is known to favor a disturbed bone metabolism. Upward arrow indicates upregulation while downward arrow indicates downregulation.

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
