# Peer review of "New Therapeutics Targeting Arterial Media Calcification: Friend or Foe for Bone Mineralization?"

_metabolites, 2022, doi:10.3390/metabo12040327_

Round 1

Reviewer 1 Report

I think this is a well organized review about targeting vascular calcification. I learned a lot from this manuscript. The authors discuss new treatment modalities targeting vascular tunica media calcification.  As the authors point out, no method has been established to control calcification in humans. Since this is a review paper, there are no particularly interesting points and it is not original. I think the paper is well written. I think a conclusion would be good.

Is there any problem facing nanoparticles?

In the nutrients section, you can mention the anti-calcification effects of magnesium and eicosapentaenoic acid.

Author Response

Point 1:  I think a conclusion would be good.

Response 1: Additional information was included in the conclusion paragraph, at line 463. 

Point 2: Is there any problem facing nanoparticles?

Response 2: Indeed, various issues should be taken into account when optimizing/using a nanoparticle drug delivery system. In example, controlling nanoparticle size, charge and distribution can be tricky as these parameters may importantly influence and determine the in vivo fate of the nanoparticles including systemic distribution, cellular uptake and circulation lifetime. We have included this information at lines 418-425.

Point 3: In the nutrients section, you can mention the anti-calcification effects of magnesium and eicosapentaenoic acid.

Response 3: We appreciate the reviewer’s relevant suggestion and have included a paragraph on the beneficial effects of magnesium and eicosapentaenoic acid on inhibiting arterial calcification (lines 382-393).

Reviewer 2 Report

In this review article, Astrid Van den Branden and co-authors present the therapeutic targets that have recently emerged to block vascular calcification. This is indeed a really important pathological process in need of pharmacological inhibitors. This review is important and timely. It is well-written and clear. There are however some concerns that should be addressed:

  • In the abstract and several locations in the text, authors write that vascular calcification is a complication of osteoporosis. Although there is an positive correlation between the progression of osteoporosis and the development of vascular calcification, osteoporosis is usually not considered (or at least presented) as a cause of vascular calcification, as can be type 2 diabetes and chronic kidney disease. Moreover, vascular calcification associated with osteoporosis in individuals without CKD and diabetes mainly develops in the intima, within atherosclerotic plaques. Authors should replace osteoporosis with atherosclerosis, and present how atherosclerotic plaques calcify.
  • A first chapter presenting how arteries calcify would be very useful. Authors could indicate that media calcification in CKD and type 2 diabetes has been reported to involve a cartilaginous metaplasia, whereas this is not the case in atherosclerotic plaques, where only intramembranous ossification has sometimes been reported. This presentation should indicate whether differences have been reported among animal models and humans. The same should been done for GACI, PXE and ACDC: does calcification develop through a process resembling ossification? This is important since blocking VSMC trans-differentiation is a therapeutic option to block vascular calcification, and since there is a competition between osteoblast and chondrocyte differentiation from MSCs (some factors, including Wnt family members, inhibiting osteoblast commitment favoring chondrocyte commitment and vice versa). Furthermore, plaque calcification develops through VSMC transdifferentiation in mice, but not in human carotids or coronaries, where the induction of calcification seems to be independent from VSMC transdifferentiation. A thorough presentation of the different mechanisms involved as a function of location, disease and species is necessary, because the calcification mechanisms will condition the therapeutic approaches to be considered.
  • The recently published articles using inositol hexakisphosphate are really interesting, and should be presented more in depth.
  • There is more evidence of a clinical association between calcification with necrosis than with apoptosis. Apoptotic bodies may not induce calcification before their membrane rupture, and therefore before secondary necrosis occurs. Since caspase inhibitors are often used in vitro to trigger necroptosis while inhibiting apoptosis, in some situations they might trigger calcification. This could be discussed.

Round 2

Reviewer 2 Report

The authors have satisfyingly answered the reviewer's comments.